# Dual Doping in Precious Metal Oxides: Accelerating Acidic Oxygen Evolution Reaction

**DOI:** 10.3390/ijms26041582

**Published:** 2025-02-13

**Authors:** Guoxin Ma, Fei Wang, Rui Jin, Bingrong Guo, Haohao Huo, Yulong Dai, Zhe Liu, Jia Liu, Siwei Li

**Affiliations:** 1Institute of Industrial Catalysis, School of Chemical Engineering and Technology, Xi’an Jiaotong University, Xi’an 710049, China; maguoxin@stu.xjtu.edu.cn (G.M.); 2236220767xjtu.stu@stu.xjtu.edu.cn (F.W.); 13403486168@163.com (R.J.); guobr@stu.xjtu.edu.cn (B.G.); hhh15343569513@163.com (H.H.); whitecat@stu.xjtu.edu.cn (Y.D.); 2Institute of Neuroscience, Translational Medicine Institute, Xi’an Jiaotong University Health Science Center, Xi’an 710061, China; liuzhe@xjtu.edu.cn; 3Department of Physiology and Pathophysiology, School of Basic Medical Sciences, Xi’an Jiaotong University Health Science Center, Xi’an 710061, China; 4Instrumental Analysis Center, Xi’an Jiaotong University, Xi’an 710049, China; liujia6j@xjtu.edu.cn

**Keywords:** electrocatalysis, acidic oxygen evolution reaction, dual doping, precious metal oxides

## Abstract

Developing a highly active and stable catalyst for acidic oxygen evolution reactions (OERs), the key half-reaction for proton exchange membrane water electrolysis, has been one of the most cutting-edge topics in electrocatalysis. A dual-doping strategy optimizes the catalyst electronic environment, modifies the coordination environment, generates vacancies, and introduces strain effects through the synergistic effect of two elements to achieve high catalytic performance. In this review, we summarize the progress of dual doping in RuO_2_ or IrO_2_ for acidic OERs. The three main mechanisms of OERs are dicussed firstly, followed by a detailed examination of the development history of dual-doping catalysts, from experimentally driven dual-doping systems to machine learning (ML) and theoretical screening of dual-doping systems. Lastly, we provide a summary of the remaining challenges and future prospects, offering valuable insights into dual doping for acidic OERs.

## 1. Introduction

As global energy demands surge and fossil fuel consumption intensifies, energy scarcity and environmental degradation are worsening, with a drastic rise in carbon dioxide emissions further intensifying climate change [1,2]. Hydrogen, boasting high energy density, renewability, and a pollution-free profile, emerges as a pivotal energy carrier for curbing fossil fuel use and attaining “zero-carbon” objectives [3,4,5]. Compared to traditional hydrogen production from fossil fuels, electrochemical water splitting offers a sustainable alternative. It leverages renewable electricity to split water into hydrogen and oxygen, yielding high-purity hydrogen without carbon dioxide emissions, thus garnering significant attention [6,7,8]. Unlike conventional alkaline water electrolysis (AWE), proton exchange membrane water electrolysis (PEMWE) exhibits high proton conductivity, minimal gas crossover, and superior voltage efficiency, showcasing immense promise. However, the oxygen evolution reaction (OER) of PEMWE demands a strongly acidic condition and entails a sluggish four-electron transfer, hampered by high oxidation potential [9,10,11,12]. Presently, precious metal Ir and Ru oxides reign as the gold standard for acidic OER electrocatalysis [13,14,15]. IrO_2_ and RuO_2_ have complementary strengths and weaknesses. IrO_2_ offers superior stability but lower activity, while RuO_2_ provides higher activity but is less stable, underscoring the limitations of both materials. Consequently, researchers are pressed to devise innovative strategies that harmonize the activity and stability of Ir/Ru-based catalysts, catalyzing the broader application of PEMWE.

Doping with heteroatoms is a straightforward and effective approach that can precisely modulate the electronic structure, alter the coordination environment, or introduce vacancies, thereby reducing the quantity of precious metals and significantly boosting the activity of the OER [16,17,18,19]. While single-element doping is a straightforward option, it may exacerbate structural defects and compromise catalyst stability during long-term electrolysis processes. In contrast, dual doping, which involves the use of two elements, represents a more advanced approach. It has been widely applied in various electrocatalytic reactions, such as the hydrogen evolution reaction [20,21], urea oxidation reaction [22], and CO_2_ reduction reaction [23], and has demonstrated substantial potential for improving catalyst performance. For acidic OERs, by utilizing the synergistic effect of two elements, dual doping can significantly enhance the activity and stability of the catalyst, while greatly reducing the required amount of precious metals [24,25,26]. Nevertheless, the increased complexity of doping components and quantities complicates the optimization process and poses significant research challenges. Therefore, the careful selection of doping elements and their concentrations is crucial for optimizing catalyst performance throughout the entire design, synthesis, and application stages. Researchers have made significant progress in the dual doping of precious metal oxides (such as RuO_2_ or IrO_2_), starting with catalyst preparation guided by experimental appraoches [27,28,29,30]. In the subsequent stages, they have developed catalysts with well-defined components and precise contents, following machine learning (ML) and theoretical screening of the catalyst system [25,26,31]. However, to the best of our knowledge, there has not been a review focusing on this emerging field.

In this review, we summarize the significant progress made in the area of dual-doping strategies for RuO_2_ or IrO_2_ in acidic OER. The article first introduces the three main mechanisms of OER. Next, we discuss in detail the latest advancements in dual doping of RuO_2_ or IrO_2_ catalysts, starting with preparation through experimentally driven approaches and later employing ML or theoretical screening approaches. Additionally, the potential of high-entropy oxides to enhance catalyst performance is explored. Finally, we outline the challenges and potential opportunities for the dual doping of precious metal oxides.

## 2. The Mechanism of OERs

As we all know, the activity of a catalyst is closely tied to its reaction mechanism. Gaining a deep understanding of the mechanisms behind the acidic OER is essential for guiding the development of high-performance catalysts. The OER involves a four-electron transfer and is characterized by slow kinetics [32]. Especially under acidic conditions, the reaction kinetics should be significantly enhanced. Currently, there are three main mechanisms in acidic solutions (Figure 1): adsorbate evolution mechanism (AEM), lattice oxygen mechanism (LOM), and oxide path mechanism (OPM) [13]. The AEM, LOM, and OPM possess distinct catalytic mechanisms, endowing them with unique catalytic properties, which will be further discussed in the following sections.

### 2.1. Adsorbate Evolution Mechanism (AEM)

In the adsorbate evolution mechanism (AEM), the catalytic pathway unfolds at the catalyst active center, where oxygen molecules are produced from water molecules in the electrolyte via a sequence of reactions. As shown in Figure 1A, each step of this process entails the injection of one proton into the electrolyte and combines with one electron on the electrode [36,37]. Initially, water decomposes on a metal (M) site to form adsorbed *OH (M-OH). Subsequently, the adsorbed *OH then loses a proton and gets oxidized to *O (M-O). Then, the oxygen atom radical engages with H_2_O to form the intermediate *OOH (M-OOH). The rate-determining step (RDS) is typically the formation of the *OOH intermediate, which is widely recognized as the bottleneck. This is because the formation of *OOH requires overcoming a high free energy barrier, and volcano plot analysis shows that the adsorption energy ratio of *OOH to *OH intermediates lead to this step being the rate-limiting one [38,39]. Likethe study of Ru/TiO_x_ catalysts [33], the Gibbs free energy diagram for the AEM pathway confirms that the formation of the *OOH intermediate is the RDS (Figure 1D). This intermediate eventually releases an O_2_ molecule, thereby regenerating the M site. The interaction between the active site and intermediates affects the OER overpotential. Optimal binding strength is key for high-performance catalysts, as too strong or too weak binding can limit overpotential [40]. According to the traditional AEM, the OER yields several adsorbed species, like *O, *OOH, and *OH. The linear relationship between the adsorption of *OH and *OOH dictates a minimum theoretical overpotential of approximately 0.37 V [38,41], indicating the difficulty for AEM-based electrocatalysts to enhance activity by surpassing this threshold. However, some recently reported electrocatalysts have exhibited OER activity beyond the theoretical limit [42,43,44], indicating the presence of other reaction mechanisms. Therefore, the lattice oxygen mechanism (LOM) has been proposed as an alternative pathway to address the surface kinetic evolution issues and to explain the inherent overpotential barriers in the AEM [45].

### 2.2. Lattice Oxygen Mechanism (LOM)

In the lattice oxygen mechanism (LOM), the catalyst lattice oxygen is converted into O_2_ through self-oxidation, effectively addressing overpotential issues and advancing efficient OER catalysts [46]. In Figure 1B, the LOM pathway starts with the formation of *O and *OH species, similar to the AEM process. Subsequently, one of the two lattice oxygen atoms bonded to the metal site migrates to another oxygen atom, directly forming an O-O bond, and lattice oxygen engages in the reversible production of O_2_ via surface oxygen vacancies. The RDS in LOM is the coupling of surface-adsorbed *O with lattice oxygen to form O_2_, which requires breaking metal–oxygen bonds [47]. This process bypasses the *OOH formation in AEM and highlights the role of lattice oxygen participation. In the study of Cr_0.6_Ru_0.4_O_2_ catalysts, the Gibbs free energy diagram shows a lower energy barrier for the rate-limiting step compared to the AEM (Figure 1E), highlighting the kinetic advantages of the LOM [34]. As a result, the LOM mechanism breaks through scaling limitations and significantly boosts catalytic activity [48,49]. In the AEM, all oxygen atoms’ generated O_2_ originatesfrom adsorbed H_2_O. Upon adsorption of oxygen atoms onto the catalyst surface, electron transfer and conformational changes take place, leading to the formation of active sites. For the LOM, the oxygen atoms in the generated O_2_ are either wholly or partially derived from the catalyst lattice oxygen. The interplay between these lattice oxygen sites and the surface active sites on the catalyst forms the active centers that drive the catalytic activity. Overall, the LOM mechanism is kinetically advantageous compared to the AEM mechanism. However, due to the oxygen vacancies generated by LOM, they may not be filled in subsequent cycles, resulting in unstable crystal structures and severe metal dissolution [13,45,50].

### 2.3. Oxide Pathway Mechanism (OPM)

In addition to the AEM and LOM mechanisms, a novel OER pathway has proposed: the oxide pathway mechanism (OPM) [35,51]. This mechanism is particularly beneficial for designing high-performance OER electrocatalysts, as it utilizes only *O and *OH as intermediates and enables the direct coupling of *O and *O to form O_2_ (Figure 1C). In the study of Mn_0.2_Ru_0.8_O_2_ catalysts, the Gibbs free energy diagram for the OPM pathway illustrates the energy changes associated with the direct coupling of *O and *O to form O_2_ (Figure 1F) [35]. This process has a relatively low energy barrier, indicating the potential for efficient OERs. The RDS is the O-O bond formation via coupling of adjacent oxygen species, influenced by the catalyst coordination environment [52,53]. The efficiency of this step depends on the control of metal spacing, coordination number, and defect structures. This process avoids the formation of oxygen vacancy defects and additional intermediates, such as *OOH. Therefore, this method requires a more precise geometric configuration of the catalyst active sites [54,55]. Symmetric bimetallic sites with optimal atomic distances are expected to promote the O-O coupling of radicals, achieving this with a low energy barrier.

The above three mechanisms offer distinct perspectives on how OER electrocatalysts work. To determine the active mechanism for a specific catalyst in OER electrocatalysis, a combination of electrochemical measurements, in situ/operando characterization, and theoretical calculations is essential. These methods provide insights into the active sites, electron transfer, catalyst surface, and oxide species formation, helping to identify whether the dominant pathway is the adsorbate evolution mechanism AEM, lattice oxygen mechanism LOM, or OPM [56,57]. Despite advancements, the mechanism of OER electrocatalysis continues to be a hot research topic, with numerous challenges and questions still requiring solutions.

## 3. Dual Doping of RuO_2_ or IrO_2_ for Acidic Oxygen Evolution Reaction

The incorporation of a single metal or non-metal dopant can influence the properties of RuO_2_ or IrO_2_ through different mechanisms, leading to enhanced catalytic activity and/or durability towards OERs in acidic condition [43,58,59,60,61,62]. Considering dual doping as a natural approach allows for the combination of the advantages of each dopant. Dual doping can significantly improve the activity and stability of catalysts by utilizing the synergistic effect of two elements, while significantly reducing the amount of precious metals. The development of dual doping has evolved from experimentally driven methods to other approaches involving machine learning (ML) and theoretical screening. Table 1 provides a comprehensive overview of various catalysts that incorporate dual doping in RuO_2_ or IrO_2_. It is worth noting that while the focus of this review is on dual-doped RuO_2_ or IrO_2_, there are also many works on RuIrO_x_ doped by other elements individually. These single-doped systems have their own significance and contributions but are not the primary focus of our review. However, the addition of dopants can increase the complexity of the catalytic system, hence the effects of dual doping are discussed in this section.

### 3.1. Experimental Driven Dual-Doping Systems

In the early stage, there are some trial-and-error works reporting a dual-doped RuO_2_ or IrO_2_ catalyst by using frequently used elements like Co-Ni [63,64,65]. A milestone of dual-doped catalysts for acidic OERs was created by Jin and Zhang et al., in which they designed a tantalum (Ta) and thule (Tm) co-doped IrO_2_ catalyst with abundant grain boundaries through a fast pyrolysis method (Figure 2A)** [27]**. With the optimal content of dopant, the Ta_0.1_Tm_0.1_Ir_0.8_O_2−δ_ exhibited an extremely low overpotential of 198 mV at 10 mA/cm^2^, much higher than that of the mono-doped catalyst (Figure 2B). It is demonstrated by X-ray photoelectron spectroscopy (XPS) and XAFS that the co-introduction of Ta and Tm further reduce the valence state of Ir (Figure 2C). Grain boundary (GB) plays similar role in the modulation of electronic structure, and it also lowers the bond length of Ir-O. Density functional theory (DFT) calculations tell that the synergy of doping and GB results in a proper p-band center and futhers the lowest value of ΔG_O_ − ΔG_OH_, which is responsible for the outstanding activity of the Ta_0.1_Tm_0.1_Ir_0.8_O_2−δ_ with GB (Figure 2D). In this work, the authors have developed an in-depth understanding of the double-doping effect, explaining how the two doping elements affect the electronic structure of Ir and further the OER performance. Moreover, Sun and Xu et al. synthesized the tungsten (W) and stannum (Sn) co-doped IrO_x_ (Ir_0.7_W_0.2_Sn_0.1_O_x_) by sol-gel method, exhibiting a low overpotential of 236 mV (10 mA/cm^2^) [28]. The XPS tells that the co-doping of W and Sn effectively stabilizes the valence state of Ir blow 4+, thus avoiding excessive oxidation. DFT calculations show that the co-doping of W and Sn shifts the d-band center of Ir to the Fermi level, thereby optimizing the binding energy of reaction intermediates and reducing the energy barrier. In addition to IrO_x_, this research team also selected molybdenum (Mo) and cerium (Ce), which have large ionic radii, for doping into the RuO_2_ (Ru_3_MoCeO_x_) to regulate the electronic structure around the Ru-O bond [66]. Based on the XPS results, the Mo and Ce act as electron donors to transfer electrons to Ru, effectively enhancing the activity and solubility resistance of its active sites. Similarly, dual dopants such as Pt-La [67] and Mn-Fe [68] have also been reported to enhance the activity and durability of RuO_2_ or IrO_2_ by modulating the electronics structure.

There are also dual-doped catalysts trying to deal with the problem of the low durability of RuO_2_ for OERs in acid. The release of oxygen from the lattice of the electrocatalyst during the OER process leads to the generation of oxygen vacancies (V_O_), which in turn accelerates the excessive oxidation of exposed Ru. Based on this, Zhang et al. co-doped W and erbium (Er) into the lattice of RuO_2_ to change its electronic structure and avoid excessive oxidation of Ru. [24] By shifting the center of the O 2p band downwards, the formation energy (ΔG) of V_O_ in W_m_Er_n_Ru_1−m−n_O_2−δ_ significantly increased, preventing the formation of soluble high valence Ru^x>4^ (Figure 3A,B). The results show that the representative W_0.2_Er_0.7_O_2−δ_ has a low overpotential of 168 mV, can obtain 10 mA/cm^2^, and is stable for at least 500 h (Figure 3C). It can also be applied as an anode catalyst in acidic PEM, with a high current of 100 mA/cm^2^ and a duration of over 120 h. The other problem is that during the OER process, the redeposition rate of high valence state Ru^n>4+^ is slower than its dissolution rate, which limits the performance of OERs [76]. Therefore, Chai and co-workers proposed a novel self-catalyzed redeposition strategy based on dual anchoring driven by cobalt (Co) and zinc (Zn), which prolongs the acidic OER lifetime by re-depositing leached run into active Ru(OH)_6_^2−^ [69]. Co and Zn stabilize Ru by supplying electrons, thereby reducing its valence state and reinforcing the Ru-O bond (first anchoring) (Figure 3D,E). Subsequently, a directed electron flow from Co to Ru triggers the self-catalytic redeposition of Ru^n>4+^ (second anchoring), establishing a new dissolution–redeposition equilibrium for Ru (Figure 3F,G). Moreover, this process altered the length of Ru-O bonds and the defect content, thereby reducing the adsorption energy of oxygen-containing intermediates, achieving optimal activity, and addressing the long-term activity and stability issues of RuO_2_ in acidic electrolytes. In addition to the effect on the durability of OERs, it is also interesting to see that the dual-doping strategy of W-Si may also enhance the poor hydrogen evolution reaction (HER) activity of RuO_2_, leading to a bi-functional catalyst for water splitting in acid [70]. Additionally, Lu and co-workers synergistically doped Cr and B elements into RuO_2_ nanofibers (NF), inducing structural distortion in RuO_2_ [71]. The authors successfully constructed Ru-B-Cr active sites, which effectively activated the Ru sites and enhanced both HER and OER activities.

Furthermore, a deeper understanding of the role of single-component doping in catalyst design can provide valuable insights into the synergistic effects of dual doping in high-performance OER catalysts. Recently, Li et al. prepared a Zn, W co-doped Ru_3_Zn_0.85_W_0.15_O_x_ (RZW) catalyst by sol-gel method and showed a low overpotential of 200 mV and remarkable stability over 4000 h at 10 mA/cm^2^ [73]. The incorporation of high electronegativity W helps to effectively capture the electrons released by sacrificial Zn during the OER process and then transfer them to the Ru site. The enhanced electron density within the stable Ru-O-W motif significantly improves the anti-peroxidation performance of Ru active sites. This study emphasizes the key role of metal doping in regulating the electronic structure of the OER catalyst during operation. Moreover, Liu et al. synthesized rutile InSnRuO_2_ oxide with O_v_ electron polarons using a temperature-controllable pyrolysis-triggered isomorphic substitution strategy, with trivalent indium (In) species as O_v_ generators and adjacent metal ions, such as Sn or Ru, as electron donors [29]. The formation of high-density polarons is attributed to the low oxidation valence In inducing a large number of positively charged O_v_ species through charge compensation. These O_v_ species can capture free electrons from adjacent metal ions in the form of ordered and asymmetric In-O_v_-Ru-O-Sn substructures. The density of polarons can be easily adjusted by the content of trivalent In in tetravalent rutile oxide. The state-of-the-art InSnRuO_2_ exhibits an ultra-low overpotential of 183 mV at 10 mA/cm^2^ and can operate for 200 h at 50 mA/cm^2^ at a low and stable battery potential (1.56 V). The significant improvement in performance is related to the synergistic effect of the In-O_v_-Ru-O-Sn substructure stabilized by dense polarons. Additionally, Li and co-workers proposed an “elastic electron transfer” strategy, which uses strontium (Sr) and chromium (Cr) co-doping to optimize the electronic structure of Ru in both directions in order to achieve high-activity and persistent OERs [30]. The incorporation of electron-withdrawing Sr increases the oxidation state of Ru and activates Ru sites. Cr acts as an electron buffer, providing electrons to Ru in the presence of Sr and absorbing excess electrons from Sr leaching during the OER process. The bidirectional regulation property of Cr prevents the peroxidation of Ru and maintains the high oxidation state of Ru during the OER process (Figure 4A,B). The optimal Ru_3_Cr_1_Sr_0.175_ has an overpotential of 214 mV at 10 mA/cm^2^ and an extended stability of 300 h at 10 mA/cm^2^. Similarly, Lee et al. enhanced the activity of RuO_2_ by introducing vanadium (V) and improved its stability by incorporating Cr [74]. The above works propose a flexible and effective strategy for bidirectional manipulation of the Ru electronic structure through multi-metal dual doping, providing guidance for designing efficient acidic OER catalysts.

In addition to the classical dual-doping strategy, the combination of interstitial atom and dopant has also been reported to facilitate the OER process over RuO_2_. Huang et al. initially prepared PtRuSe hollow nanospheres (HNSs) via the hydrothermal method, subsequently mixed them with carbon powder, and then calcined the mixture to produce platinum (Pt)- or selenium (Se)-doped RuO_2_ hollow nanospheres (SS Pt-RuO_2_ HNSs) with interstitial carbon (C) as highly active and stable electrocatalysts. During this process, C is trapped in the interstices, Pt replaces isolated points, and Se partially substitutes for Ru sites as shown in Figure 4C [72]. The prepared catalyst can also be used for whole water splitting and exhibits good performance. It is worth noting that SS Pt-RuO_2_ HNSs exhibit good stability during continuous operation at 100 mA/cm^2^ for 100 h in a polymer electrolyte membrane electrolysis cell. Detailed experiments have shown that the C-O bond formed by gap C extends the Ru-O and Pt-O bonds (Figure 4D,E). Additionally, inspired by the literature on RuO_2_ composites with carbon materials [77,78], exploring the potential of combining RuO_2_ or IrO_2_ with M-N-C composites appears promising for enhancing catalyst performance.

### 3.2. Machine Learning and Theoretical Screening of Dual-Doping Systems

The above articles are primarily guided by experimentally driven approaches, and some successful dual-doped RuO_2_ or IrO_2_ systems have been reported. However, the question of how to select the appropriate two dopants remains unanswered. It is reasonable to speculate that a significant amount of experimentation is required to identify the suitable dopants. Machine learning (ML) has emerged as a powerful tool for screening suitable catalysts in electrocatalysis by efficiently predicting catalyst performance metrics, such as activity, stability, and selectivity based on material descriptors [79]. Sargent et al. successfully screened 2070 oxides using a machine learning-assisted computational process and used metal oxygen covalence as a secondary screening descriptor to predict the volume stability of the oxides and evaluate their electrochemical stability (Figure 5A,B)** [31]**. By combining the real calculation descriptor screening method with experiments, it was determined that Ru_0.6_Cr_0.2_Ti_0.2_O_2_ is a candidate material with high durability. Titanium (Ti) increases the metal–oxygen covalence, which is a potential pathway to increase stability, while Cr reduces the energy barrier that determines the rate of *OOH formation. As demonstrated by experiments, the Ru_0.6_Cr_0.2_Ti_0.2_O_2_ prepared by sol-gel method provides an overpotential of 267 mV at 100 mA/cm^2^ and operates for more than 200 h at this current density, with a rate of overpotential increase of 25 μV/h (Figure 5C). In our view, this study can effectively identify efficient and stable OER catalysts using ML-assisted approaches, with experimental validation to confirm their performance. The descriptors employed in ML are crucial for the screening process and can also be extended to facilitate the design of more complex multi-doped precious metal oxides.

Moreover, theoretical screening using DFT calculations has been employed to predict the catalytic performance of these systems [80]. DFT simulations have been successfully applied to study the OER catalysis. For example, a study by Sargent and co-workers also employed a dual-modulation strategy to suppress LOM and adjust the electronic structure of active Ru sites, introducing Sr and Ir into RuO_2_ (SrRuIr) [25]. The authors first predicted the OER activity and stability of the SrRuIr ternary system using DFT and determined the ratio of Sr and Ir (Figure 5D). Then, by sol-gel method synthesizing SrRuIr oxide electrocatalysts, SrRuIr achieved an overpotential of 190 mV at 10 mA/cm^2^ and remained below 225 mV after 1500 h of operation. The EXAFS and in situ differential electrochemical mass spectrometry (DEMS) measurements indicate that the participation of lattice oxygen in the OER process is inhibited by local interactions in the Ru-O-Ir structure, providing insights into how to improve stability (Figure 5E,F). Moreover, in situ XAS and DFT calculations indicate that the doping of Sr and Ir optimizes the binding energy of oxo-intermediates on high-valence Ru sites. In addition, Xiang et al. improved the activity and stability of RuO_2_ OERs by increasing the (110) lattice spacing through modulation strategy [26]. Firstly, establish theoretical activity and stability evaluation criteria and then screen and prepare Mo_0.15_Nb_0.05_-RuO_2_ as a potential OER catalyst through DFT calculations. Mo is doped first, followed by Nb, and RuO_2_ has the maximum (110) lattice spacing. Mo_0.15_Nb_0.05_-RuO_2_ exhibits the best OER activity, with an overpotential of only 205 mV at 10 mA/cm^2^.

A further step beyond the dual-doping strategy is to construct high-entropy oxides based on RuO_2_ or IrO_2_. For example, Huang et al. proposed a design strategy for multicomponent catalysts to address the issues of poor stability and limited catalytic activity of RuO_2_ under acidic OER conditions. The authors incorporated both acid-resistant components (Ir) and activity-enhancing components (Fe, Co, Ni) into RuO_2_ and prepared a five-element high-entropy oxide (M-RuIrFeCoNiO_2_) rich in GB through a rapid and non-equilibrium method (Figure 6A) [75]. Microscopic analysis, DFT, and DEMS with isotope labeling indicate that the incorporation of exogenous metal elements and GB can effectively alter the electronic structure and OER pathway of RuO_2_, inhibit the participation of lattice oxygen in the OER process, and thus improve its activity and stability (Figure 6B,C). Meanwhile, the synergistic effect of multiple exogenous metal elements with GB can regulate the binding energy of oxygen intermediates and accelerate the reaction kinetics of OERs. Therefore, the OER overpotential required for the prepared M-RuIrFeCoNiO_2_ catalyst to reach a current density of 10 mA/cm^2^ is only 189 mV (Figure 6D,E). The authors combine the advantages of various components and GB, breaking through the limitations of thermodynamic solubility of different metal elements.

In summary, doping RuO_2_ or IrO_2_ with two or more elements is an effective strategy to simultaneously achieve high activity and stability. Doped catalysts can be synthesized via several methods, each with its own advantages. The sol-gel method, which involves hydrolyzing and condensing metal alkoxides or salts to form a gel that is then calcined, ensures homogeneous mixing and precise control (e.g., Ru_3_MoCeO_x_). Hydrothermal/solvothermal synthesis, which reacts with metal precursors in a high-temperature/pressure solvent, produces highly crystalline materials with atomic-level doping (e.g., Ta_0.1_Tm_0.1_Ir_0.8_O_2−δ_). Co-precipitation, which precipitates metal ions together, is simple and cost-effective, yielding uniform mixed-metal oxides (e.g., CoIr-CLEO). Impregnation, which loads metal precursors onto a support, enhances stability and activity by loading active metals onto supports (e.g., Pt_0.1_La_0.1_-IrO_2_@NC). Electrochemical methods, which deposit metal ions onto a substrate, create active and stable catalysts with controlled morphologies (e.g., Co_0.2_Zn_0.8_@RuO_2_). However, establishing a clear structure–property relationship is challenging due to the complexity of these systems. The research paradigm in this field is progressively shifting from traditional experimentally driven approaches to a more rational design, informed by ML and theoretical screening. In our view, researchers may increasingly focus on exploring high-entropy oxides, as they not only enhance catalytic performance in the OER but also significantly reduce the use of precious metals, thereby lowering the costs associated with PEMWE.

## 4. Summary and Outlook

Dual doping of RuO_2_ or IrO_2_ systems confers notable benefits regarding acidic OER activity and stability. In contrast to single doping, which can boost catalytic performance to some degree yet often triggers structural defects that cap enhancements, dual doping taps into the synergy between two dopant elements. This strategy significantly cuts down on the amount of precious metals used, thus markedly boosting activity and stability. The impacts of dual doping involve modulating the electronic structure, adjusting the coordination environment, generating vacancy formation, and inducing strain effects. Through these diverse avenues, dual doping effectively bolsters and stabilizes the active sites, thereby optimizing OER performance. The evolution of dual doping catalysis has traversed a progressive journey. Initially, researchers enhanced performance through direct experimental doping strategies, such as introducing elements like Ta-Tm and W-Er into IrO_2_ or RuO_2_. This foundational approach gradually transitioned into more designed doping strategies, such as the incorporation of In-Sn and Sr-Cr into RuO_2_. Subsequently, the field advanced to employ sophisticated techniques, like machine learning and theoretical screening, targeting dopant pairs, such as Cr-Ti and Mo-Nb into RuO_2_. Ultimately, research has shifted towards exploring more complex high-entropy systems. This marks a significant leap in the complexity and sophistication of dual-doping strategies.

Although dual doping has made substantial strides in boosting the performance of acidic OERs, numerous fundamental facets remain in the experimental phase. Key areas include the application of machine learning and theoretical screening to multicomponent doping systems, revealing the dual-doping effects via in situ characterizations and extending dual-doping strategies to non-noble metal oxides.

### 4.1. Machine Learning and Theoretical Screening for Multicomponent Doping Systems

Dual-doped RuO_2_ and IrO_2_ systems offer significant advantages in acidic OER activity and stability due to synergistic effects. These systems also allow for more flexible composition adjustments, paving the way for the development of multicomponent doping or high-entropy Ir/Ru oxides, which aim to further optimize acidic OER performance by reducing reliance on precious metals while simultaneously improving both activity and stability. The multicomponent doping system of acid OER catalysts can achieve multidimensional control of the catalyst electronic structure and reaction pathway through the synergistic effect of multiple elements, demonstrating great potential for application [81]. However, its complexity poses challenges for experimental design and material screening.

The combination of machine learning (ML) and theoretical screening techniques (such as DFT calculations) provides new ideas for studying multicomponent doping systems [82,83]. ML can extract information from large amounts of data, establish a correlation model between catalyst performance and material characteristics, and efficiently screen catalyst combinations with excellent performance. Meanwhile, theoretical screening, such as DFT, can simulate the interaction between doped elements and catalyst substrates at the atomic level and predict the effects of different doping combinations on catalyst performance. With the improvement of computing power and algorithm optimization, this combination strategy will provide important support for the development of high-performance acidic OER catalysts.

### 4.2. Revealing the Dual-Doping Effects via In Situ Characterizations

Currently, most of the doping effects for RuO_2_ and IrO_2_ lie in the modulation of the electronic structure, and the influence of the change of electron state density at the active sites on the adsorption/desorption of reaction intermediates is reflected by DFT calculations. However, DFT calculations are limited in their ability to fully simulate real reaction processes, particularly for doped systems with dynamic charge transfer [84]. Additionally, the OER process may involve dynamic changes in Ru/Ir-O or Ru/Ir-O-M bond lengths as the applied voltage increases. Therefore, the development of advanced in situ characterization techniques, such as XPS, synchrotron X-ray absorption spectroscopy (XAS), attenuated total reflection surface-enhanced infrared absorption spectroscopy (ATR-SEIRAS), transmission electron microscope (TEM), and Raman spectroscopy, is essential. For example, in the study of the Cr_0.6_Ru_0.4_O_2_ catalyst for acidic OERs, in situ Raman spectroscopy, in situ ATR-SEIRAS, XPS, XAS, and DFT calculations were used together [34]. These techniques showed that Cr doping optimizes the Ru-O covalency, enhances the catalyst stability and activity, and promotes the direct oxygen radical coupling mechanism for an efficient OER in acidic conditions. These techniques can monitor the chemical composition of electrocatalysts, changes in the oxidation states of active sites, nanoscale structural changes, surface dissolution, and real-time atomic-scale variations. These will facilitate the observation of catalyst surface evolution and enhance our understanding of the effects of doping on the reaction process.

### 4.3. Expanding Dual-Doping Strategies to Non-Noble Metal Oxides

Given the scarcity and high cost of noble metals such as Ru and Ir, it is of great strategic significance to replace them with cost-effective non-noble metals for long-term cost reduction and resource conservation [85,86]. Co-based and Mn-based catalysts have attracted considerable research interest, primarily due to their low cost, high acid resistance, and catalytic performance. Nevertheless, to satisfy the stringent demands of PEMWE, it is imperative to further elevate their performance. Some advancements have been made in single-element doping of Co-based and Mn-based spinel catalysts, such as Er-Co_3_O_4_ [87], Co_1.8_Ga_1.2_O_4_ [88], Ir-MnO_x_ [89], and Ru_0.1_Mn_0.9_O_x_ [90]. However, research on the single doping strategy for these specific compounds remains relatively limited. Extending the dual-doping strategy from noble metal oxides to non-noble metal oxides, such as Co-based and Mn-based oxides, is anticipated to markedly enhance the electrocatalytic performance. These approaches offer a promising avenue to improve the catalytic efficiency and feasibility of non-noble metal oxides in PEMWE applications.

## Figures and Tables

**Figure 1 ijms-26-01582-f001:**
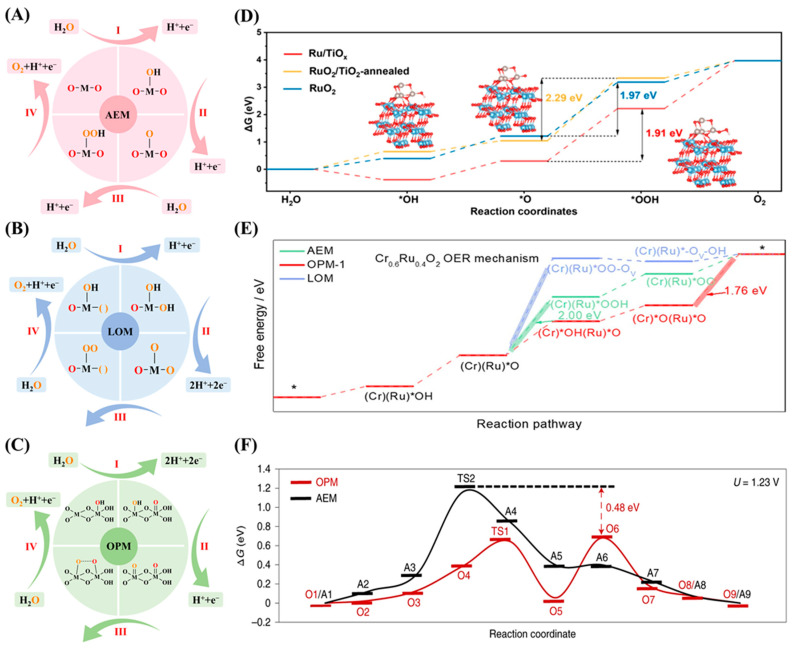
Schematic illustration of AEM (**A**), LOM (**B**), and OPM (**C**) in the OER process. (The symbol “( )” represents oxygen vacancies and M stands for metal). (**D**) Gibbs free energy diagram of the OER on Ru/TiO*_x_*, RuO_2_, and RuO_2_/TiO_2_-annealed. (The symbol “*” represents adsorption sites or active sites on the catalyst surface.) (**D**) was reproduced with permission [33]. Copyright 2025, American Chemical Society. (**E**) Free energy diagrams for OER paths on the surfaces of Cr_0.6_Ru_0.4_O_2_. (**E**) was reproduced with permission [34]. Copyright 2024, American Chemical Society. (**F**) The free energy (Δ*G*) diagrams of AEM and OPM at 1.23  V versus RHE. (**F**) was reproduced with permission [35]. Copyright 2021, Nature Publishing Group.

**Figure 2 ijms-26-01582-f002:**
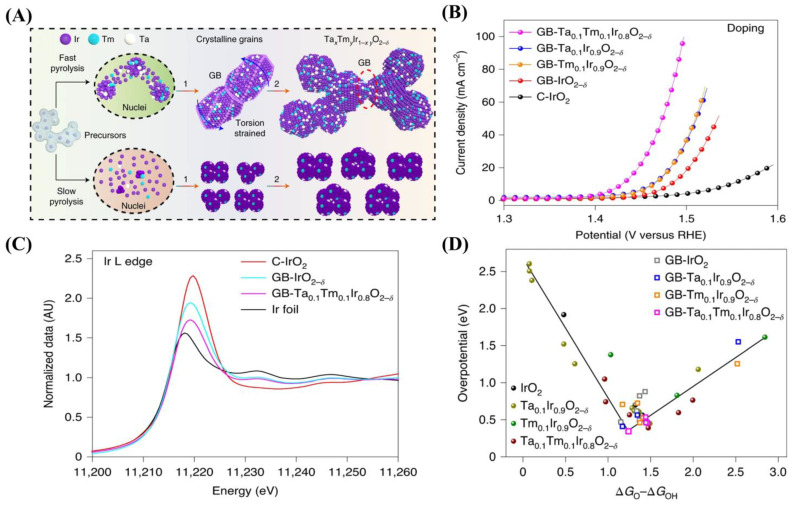
(**A**) The schematic routes for synthesizing GB-Ta_0.1_Tm_0.1_Ir_0.8_O_2−δ_ nanocatalyst. (**B**) LSV curves of different doped GB-Ta_x_TmyIr_1−x−y_O_2−δ_ samples versus C-IrO_2_. (**C**) Ir L-edge XANES spectra for Ir foil, C-IrO_2_, GB-IrO_2−δ_ and GB-Ta_0.1_Tm_0.1_Ir_0.8_O_2−δ_ nanocatalysts. (**D**) The volcano plot for the overpotential as a function of ΔG_O_ − ΔG_OH_ for various Ta_x_TmyIr_1−x−y_O_2−δ_ catalyst models through doping and strain effects. (**A**–**D**) were reproduced with permission [27]. Copyright 2021, Nature Publishing Group.

**Figure 3 ijms-26-01582-f003:**
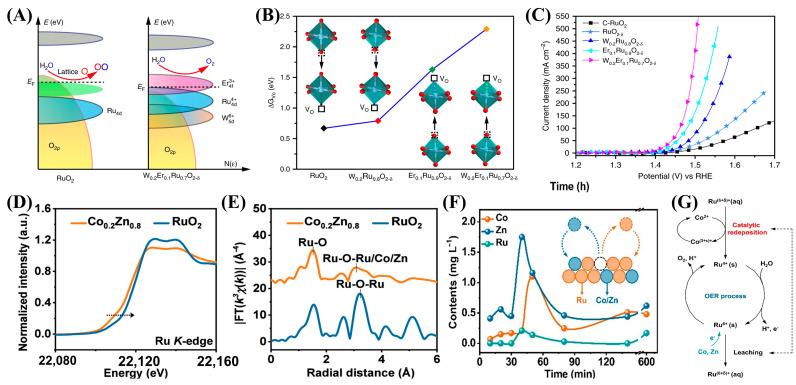
(**A**) Schematic diagrams of rigid band models for RuO_2_ and W_0.2_Er_0.1_Ru_0.7_O_2−*δ*−1_. (**B**) The calculated energy for formation of V_O_ in different positions of RuO_2_, W_0.2_Ru_0.8_O_2−*δ*−1_, Er_0.1_Ru_0.9_O_2−*δ*−1_, and W_0.2_Er_0.1_Ru_0.7_O_2−*δ*−1_. (**C**) Polarization curves. (**A**–**C**) were reproduced with permission [24]. Copyright 2020, Nature Publishing Group. (**D**) Ru k-edge XANES spectra and (**E**) Fourier transforms of k^3^-weighted EXAFS signals of RuO_2_ and Co_0.2_Zn_0.8_@RuO_2_ after OER measurement. (**F**) ICP contents of Co, Zn, and Ru in electrolyte. (**G**) Self-catalyzed redeposition mechanism of Co_0.2_Zn_0.8_@RuO_2_. (**D**–**G**) were reproduced with permission [69]. Copyright 2024, American Chemical Society.

**Figure 4 ijms-26-01582-f004:**
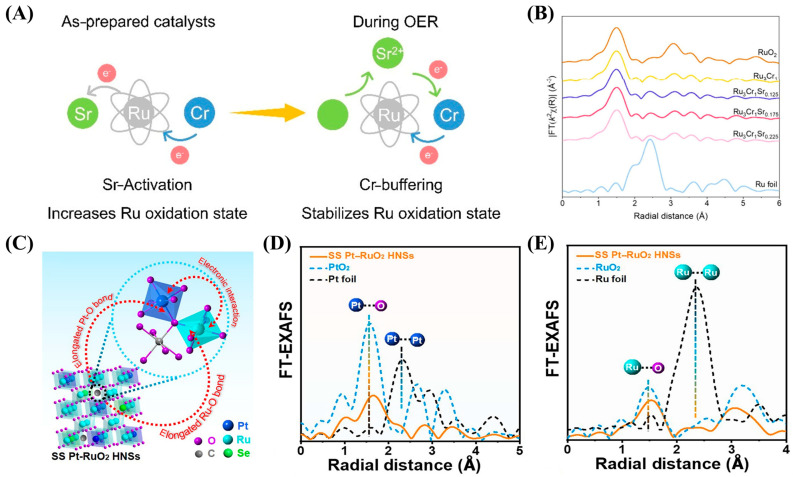
(**A**) Schematic diagram of the effects of incorporating Sr and Cr. (**B**) FT-EXAFS spectra at the Ru Kedge of as-prepared Ru_3_Cr_1_Sr_x_ catalysts, standard Ru foil, and RuO_2_. (**A**,**B**) were reproduced with permission [30]. Copyright 2024, American Chemical Society. (**C**) Structural illustration of SS PtRuO_2_ HNSs. (**D**) Pt L_3_-edge EXAFS spectra of SS Pt-RuO_2_ HNSs, PtO_2_, and Pt foil. (**E**) Ru Kedge EXAFS spectra of SS Pt-RuO_2_ HNSs, commercial RuO_2_, and Ru foil. (**C**–**E**) were reproduced with permission [72]. Copyright 2022, American Association for the Advancement of Science.

**Figure 5 ijms-26-01582-f005:**
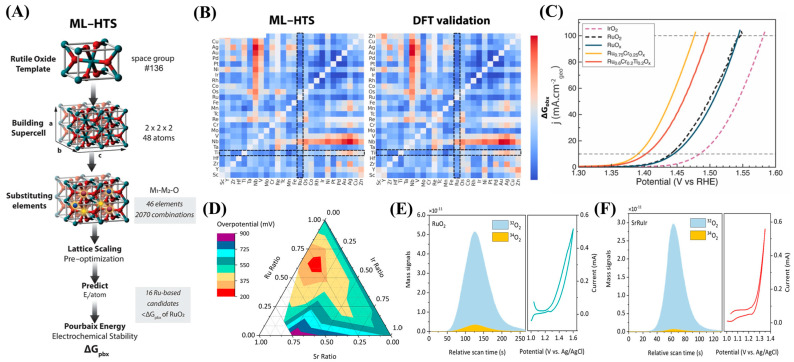
(**A**) Diagram illustrating the workflow of the machine learning high-throughput screening (ML-HTS) pipeline employed to predict the Pourbaix electrochemical stability (ΔG_pbx_) for 2070 rutile oxide structures. (**B**) CGCNN-HD heatmaps of ΔG_pbx_ vs. DFT validation stability heatmap for the subset. Highlighted in the dashed area is Ru-Ti as the most stabilizing with a nonprecious metal. (**C**) Calculated OER overpotential with different ternary compositions. The three edges of the ternary plot correspond to the ratio of two metal elements (i.e., Ir-Sr, Ir-Ru, and Sr-Ru). (**A**–**C**) were reproduced with permission [31]. Copyright 2024, American Chemical Society. (**D**) Schematic diagram of the effects of incorporating Sr and Cr. (**E**,**F**) DEMS signals of ^32^O_2_ (^16^O^16^O) and ^34^O_2_ (^16^O^18^O) from the reaction products for ^18^O-labeled SrRuIr and RuO_2_ catalysts in H_2_^18^O aqueous sulfuric acid electrolyte and corresponding CV cycles. (**D**–**F**) were reproduced with permission [25]. Copyright 2021, American Chemical Society.

**Figure 6 ijms-26-01582-f006:**
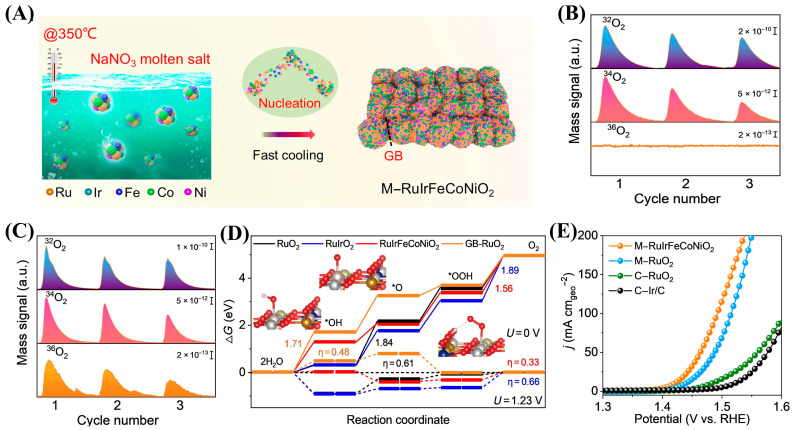
(**A**) Schematic illustration of the fast, nonequilibrium synthetic process for synthesizing M-RuIrFeCoNiO_2_, showing the generation of GB. (**B**,**C**) DEMS signals of ^32^O_2_ [^16^O^16^O, mass/charge ratio (*m*/*z*) = 32], ^34^O_2_ (^16^O^18^O, *m*/*z* = 34), and ^36^O_2_ (^18^O^18^O, *m*/*z* = 36) from the gaseous products for ^18^O-labeled M-RuIrFeCoNiO_2_ (**B**) and M-RuO_2_ (**C**) catalysts in H_2_^18^O aqueous H_2_SO_4_ electrolyte at three times during cycles in the potential range of 1.2 to 1.55 V versus RHE at a scan rate of 10 mV/s. (**D**) Free energy diagram for OERs over RuO_2_, RuIrO_2_, RuIrFeCoNiO_2_, and GB-RuO_2_ surfaces at zero cell potential (U = 0), denoted by a solid line, and at the equilibrium potential (U = 1.23 V), denoted by a dotted line. (**E**) Polarization curves of M-RuIrFeCoNiO_2_, M-RuO_2_, and commercial control catalysts (C-RuO_2_ and C-Ir/C) with iR correction. (**A**–**E**) were reproduced with permission [75]. Copyright 2023, American Association for the Advancement of Science.

**Table 1 ijms-26-01582-t001:** The comparisons of reported catalyst for dual-doped RuO_2_ or IrO_2_.

Catalyst	Electrolyte	η [mV] to 10 mA/cm^2^	Tafel Slope [mV/dec]	Stability	Ref.
Ni_0.4_Co_0.6_	0.1 M HClO_4_	/	32	5 h @1.58 V (vs. RHE)	[63]
CoIr-CL_EO_	0.5 M H_2_SO_4_	/	39.8 ± 1.8	13.5 h @1.6 V (vs. RHE)	[64]
ICN-50	0.1 M HClO_4_	285	53	20,000 s @10 mA/cm^2^	[65]
Ta_0.1_Tm_0.1_Ir_0.8_O_2−δ_	0.5 M H_2_SO_4_	198	64	500 h @10 mA/cm^2^	[27]
Ir_0.7_W_0.2_Sn_0.1_O_x_	0.5 M H_2_SO_4_	236	57	220 h @ 1 A/cm^2^	[28]
Ru_3_MoCeO_x_	0.5 M H_2_SO_4_	164	61.2	100 h @10 mA/cm^2^	[66]
Pt_0.1_La_0.1_-IrO_2_@NC	0.5 M H_2_SO_4_	205	50.9	135 h @10 mA/cm^2^	[67]
MnFeRu-90	0.1 M HClO_4_	270	41	5 h @1.58V (vs. RHE)	[68]
W_0.2_Er_0.1_Ru_0.7_O_2−δ_	0.5 M H_2_SO_4_	168	66.8	500 h @10 mA/cm^2^	[24]
Co_0.2_Zn_0.8_@RuO_2_	0.5 M H_2_SO_4_	150	49.5	500 h @100 mA/cm^2^	[69]
RuSiW	0.5 M H_2_SO_4_	142	61.3	100 h @10 mA/cm^2^	[70]
Cr, B-doped RuO_2_ NFs	0.5 M H_2_SO_4_	379 @1 A/cm^2^	58.9	188 h @ 1 A/cm^2^	[71]
SS Pt-RuO_2_ HNSs	0.5 M H_2_SO_4_	228	51	100 h @10 mA/cm^2^	[72]
Ru_3_Zn_0.85_W_0.15_O_x_	0.1 M HClO_4_	200	37.6	4000 h @10 mA/cm^2^	[73]
InSnRuO_2_	0.5 M H_2_SO_4_	183	49.78	200 h @10 mA/cm^2^	[29]
Ru_3_Cr_1_Sr_0.175_	0.1 M HClO_4_	214	40.6	300 h @10 mA/cm^2^	[30]
Ru_0.45_V_0.50_Cr_0.05_O_2_	0.1 M H_2_SO_4_	241	33.8	6 h @10 mA/cm^2^	[74]
Ru_0.6_Cr_0.2_Ti_0.2_O_x_	0.5 M H_2_SO_4_	267 @100 mA/cm^2^	58	200 h @100 mA/cm^2^	[31]
SrRuIr	0.5 M H_2_SO_4_	180 ± 5	39	1500 h @10 mA/cm^2^	[25]
Mo_0.15_Nb_0.05_-RuO_2_	0.5 M H_2_SO_4_	205	48.9	80 h @200 mA/cm^2^	[26]
M-RuIrFeCoNiO_2_	0.5 M H_2_SO_4_	189	49	500 h @ 1 A/cm^2^	[75]

## Data Availability

Not applicable.

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
