# Peer review of "Dual Doping in Precious Metal Oxides: Accelerating Acidic Oxygen Evolution Reaction"

_ijms, 2025, doi:10.3390/ijms26041582_

Round 1
Reviewer 1 Report
Comments and Suggestions for Authors
Oxygen evolution in acidic media is of significant importance in the field of electrocatalysis. Doping RuO2 and IrO2 is a potential strategy to achieve high activity and stability simultaneously. The present manuscript timely reviews the progress in dual-element-doped noble metal oxides (NMOs) for acidic OER. It discusses the reaction mechanism and further reviews the development of dual-doped NMOs from trial-and-error studies to rational design based on machine learning. Moreover, the authors provide an in-depth summary and perspective on this topic. I recommend it for acceptance for publication in IJMS after minor revisions. Listed below are my comments and suggestions.
1. Dual-doping strategy, the key point of this review, is a widely discussed topic in heterogeneous catalysis. I suggest the authors to give a brief introduction of this strategy in Introduction part to enhance the broad interests of this mini-review.
2. As a hot topic in electrocatalysis, there have been recent publications about dual-doped NMOs. For example, the W,Zn doped RuO2 system published on Angew. Chem. https://doi.org/10.1002/anie.202422707
3. There are plenty of works about RuIrOx doped by other elements, which have not been included in this review. I agree with the authors that these works are single-doped rather than dual-doped NMOs; however, they may add a statement in Part 3 to make it more rigorous.
4. Finally, and very importantly, the copyright permissions should be stated for each case clearly.
Author Response
Comments:
Oxygen evolution in acidic media is of significant importance in the field of electrocatalysis. Doping RuO2 and IrO2 is a potential strategy to achieve high activity and stability simultaneously. The present manuscript timely reviews the progress in dual-element-doped noble metal oxides (NMOs) for acidic OER. It discusses the reaction mechanism and further reviews the development of dual-doped NMOs from trial-and-error studies to rational design based on machine learning. Moreover, the authors provide an in-depth summary and perspective on this topic. I recommend it for acceptance for publication in IJMS after minor revisions. Listed below are my comments and suggestions.
Answer: Many thanks for the positive comments.
- Dual-doping strategy, the key point of this review, is a widely discussed topic in heterogeneous catalysis. I suggest the authors to give a brief introduction of this strategy in Introduction part to enhance the broad interests of this mini-review.
Answer: Thanks for this nice comment. We agree with the reviewer’s insight and rewrite the introduction of the dual-doping strategy in the Introduction section to enhance the broad interest of this mini-review. The dual-doping is a universal strategy that has been widely applied in various electrocatalytic reactions. In this revision, for hydrogen evolution reaction, we have read the paper (Small 2024, 20, (40), 2402615; Advanced Materials 2024, 36, (33), 2405970.) carefully and found these works demonstrate the effective use of the dual-doping strategy to design high-performance electrocatalysts for energy conversion. These references have been cited as [20] and [21] in the revised manuscript. For urea oxidation reaction, we have examined the studies (Journal of Colloid and Interface Science 2023, 650, 1851-1861.). This work uses a dual doping strategy to develop efficient bifunctional electrocatalysts for OER and urea oxidation reaction, promoting the enormous potential of electrocatalysts in wastewater purification and hydrogen production, cited in our manuscript as references [22]. For CO2 reduction reaction, we have reviewed the paper (Nature Communications 2022, 13, (1), 1965.). This study introduces a straightforward in situ dual-doping strategy involving cations and anions to construct highly efficient electrocatalysts for converting carbon dioxide to methanol. The findings suggest that this in situ dual-doping approach can also be applied to design other high-performance electrocatalysts. This reference is cited as [23] in the revised manuscript.
In the revised version, We rewrite the introduction to enhance its broad interest, as “While single-element doping is a straightforward option, it may exacerbate structural defects and compromise catalyst stability during long-term electrolysis processes. In contrast, dual-doping, which involves the use of two elements, represents a more advanced approach. It has been widely applied in various electrocatalytic reactions, such as the hydrogen evolution reaction [20, 21], urea oxidation reaction [22], and CO2 reduction reaction [23], and has demonstrated substantial potential for improving catalyst performance. For acidic OER, by utilizing the synergistic effect of two elements, dual doping can significantly enhance the activity and stability of the catalyst, while greatly reducing the required amount of precious metals [24-26].”.
- As a hot topic in electrocatalysis, there have been recent publications about dual-doped NMOs. For example, the W, Zn doped RuO2 system published on Angew. Chem. https://doi.org/10.1002/anie.202422707.
Answer: Thank you for your reminder. We greatly appreciate the literature information you provided. In the revised manuscript, we have added this literature in the Part 3, as “Recently, Li et al. prepared Zn, W co-doped Ru3Zn0.85W0.15Ox (RZW) catalyst by sol-gel method, and showed a low overpotential of 200 mV and remarkable stability over 4000 hours at 10 mA/cm2 [73]. The incorporation of high electronegativity W helps to effec-tively capture the electrons released by sacrificial Zn during the OER process, and then transfer them to the Ru site. The enhanced electron density within the stable Ru-O-W motif significantly improves the anti-peroxidation performance of Ru active sites. This study emphasizes the key role of metal doping in regulating the electronic structure of OER catalyst during operation.”.
- There are plenty of works about RuIrOx doped by other elements, which have not been included in this review. I agree with the authors that these works are single-doped rather than dual-doped NMOs; however, they may add a statement in Part 3 to make it more rigorous.
Answer: Thank you very much for your suggestion. We fully agree that there are indeed numerous studies on RuIrOx doped by other elements, and we appreciate your reminder. We have added a statement in Part 3 to address this issue, as “Table 1 provides a comprehensive overview of various catalysts that incorporate dual doping in RuO2 or IrO2. It is worth noting that while the focus of this review is on dual-doped RuO2 or IrO2, there are also many works on RuIrOx doped by other elements individually. These single-doped systems have their own significance and contributions, but are not the primary focus of our review. However, the addition of dopants can increase the complexity of the catalytic system, hence the effects of dual doping are discussed in this section.”. We believe this satement make our review more comprehensive and rigorous.
- Finally, and very importantly, the copyright permissions should be stated for each case clearly.
Answer: Thank you for your reminder. We ensure the copyright permissions for all figures are clearly stated in the captions.
Reviewer 2 Report
Comments and Suggestions for Authors
Please find additional comments in the attached document.

Author Response
Comments:
In this manuscript, the authors report a review of doping strategies in precious metal oxides and their improved electrochemical oxygen evolution reaction in acidic environment. Three mechanisms of OER have been discussed, including adsorbate evolution mechanism, lattice oxygen mechanism, and oxide pathway mechanism. Various catalyst examples were discussed in this report and divided into different investigation methods, such as experimental, Machine learning, and theoretical screening. Although the manuscript is in a good shape, some important information is missing. The manuscript can be considered as acceptance after some revisions. Additional comments are listed as follows:
Answer: Many thanks for the positive comments.
- In the Mechanism section, what is the rate determining step for each mechanism?
Answer: Thank you for your insightful comment. We appreciate the opportunity to clarify the rate determining steps (RDS) for each mechanism.
- Adsorbate Evolution Mechanism (AEM): The RDS is typically the formation of the *OOH intermediate. This is because the formation of *OOH requires overcoming a high free energy barrier (ChemCatChem 2011, 3, (7), 1159-1165.). Additionally, volcano plot analysis shows that the adsorption energy ratio of *OOH to *OH intermediates leads to this step being the rate-limiting one (Chemical Science 2023, 14, (39), 10644-10663.). We have added this section in the revised version as “ The rate determining step (RDS) is typically the formation of the *OOH intermediate, which is widely recognized as the bottleneck. This is because the formation of *OOH requires overcoming a high free energy barrier, and volcano plot analysis shows that the adsorption energy ratio of *OOH to *OH intermediates lead to this step being the rate-limiting one [38, 39].”.
- Lattice Oxygen Mechanism (LOM): The RDS in LOM is the coupling of surface-adsorbed *O with lattice oxygen to form O2, which requires breaking metal-oxygen bonds (Advanced Functional Materials 2024, 34, (32), 2401610; Molecules 2024, 29, (2), 537.) . This process bypasses the *OOH formation in AEM and highlights the role of lattice oxygen participation. It is revised as“ The RDS in LOM is the coupling of surface-adsorbed *O with lattice oxygen to form O2, which requires breaking metal-oxygen bonds [47]. This process bypasses the *OOH formation in AEM and highlights the role of lattice oxygen participation. In the study of Cr0.6Ru0.4O2 catalysts, the Gibbs free energy diagram shows a lower energy barrier for the rate-limiting step compared to the AEM (Figure 1E), highlighting the kinetic advantages of the LOM [34]. As a result, the LOM mechanism breaks through scaling limitations and significantly boosts catalytic activity [48, 49].”.
- Oxide Pathway Mechanism (OPM): The RDS is the O-O bond formation via coupling of adjacent oxygen species, influenced by the catalyst coordination environment (Nano-Micro Letters 2022, 14, (1), 112; Journal of Materials Chemistry A 2025, 13, (1), 587-594.). The efficiency of this step depends on the control of metal spacing, coordination number, and defect structures. We have added in the revised manuscript as “The RDS is the O-O bond formation via coupling of adjacent oxygen species, influenced by the catalyst coordination environment [52, 53]. The efficiency of this step depends on the control of metal spacing, coordination number, and defect structures.”.
- It is of interest to show example of Gibbs free energy diagrams of different mechanisms.
Answer: Thank you for your valuable suggestion. We agree that including examples of Gibbs free energy diagrams for different mechanisms would be beneficial for readers to better understand the reaction pathways and energy barriers involved in each mechanism. The examples are as follows.
- Adsorbate Evolution Mechanism (AEM): In the study of Ru/TiOx catalysts (ACS Catalysis 2025, 15, (2), 768-779.), the Gibbs free energy diagram for the AEM pathway reveals that the formation of the *OOH intermediate is the RDS of the OER process. We have added in the revised version as “ Such as the study of Ru/TiOx catalysts [33], the Gibbs free energy diagram for the AEM pathway confirms that the formation of the *OOH intermediate is the RDS (Figure 1D).”
- Lattice Oxygen Mechanism (LOM): In the study of Cr0.6Ru0.4O2 catalysts (Journal of the American Chemical Society 2024, 146, (46), 32049-32058.), the Gibbs free energy diagram shows a lower energy barrier for the rate-limiting step compared to the AEM, highlighting the kinetic advantages of the LOM. It is revised as“ In the study of Cr0.6Ru0.4O2 catalysts, the Gibbs free energy diagram shows a lower energy barrier for the rate-limiting step compared to the AEM (Figure 1E), highlighting the kinetic advantages of the LOM [34].”.
- Oxide Pathway Mechanism (OPM): In the study of Mn0.2Ru0.8O2 catalysts, the Gibbs free energy diagram for the OPM pathway illustrates the energy changes associated with the direct coupling of *O and *O to form Oâ‚‚. This process has a relatively low energy barrier, indicating the potential for efficient OER. We have added in the revised manuscript as “In the study of Mn0.2Ru0.8O2 catalysts, the Gibbs free energy diagram for the OPM pathway illustrates the energy changes associated with the direct coupling of *O and *O to form O2 (Figure 1F) [35].”.
- How to determine which mechanism is active for a specific catalyst?
Answer: Thank you for your valuable suggestion. We have added a detailed section on how to determine the mechanism for a specific catalyst, as“ To determine the active mechanism for a specific catalyst in OER electrocatalysis, a combination of electrochemical measurements, in-situ/operando characterization, and theoretical calculations is essential. These methods provide insights into the active sites, electron transfer, catalyst surface, and oxide species formation, helping to identify whether the dominant pathway is the adsorbate evolution mechanism AEM, lattice oxygen mechanism LOM, or OPM [56,57].”.
- How to synthesize doped catalyst and what is the advantage for different methods?
Answer: Thank you for your insightful comment. The synthesis of doped catalysts can be achieved through various methods, each offering distinct advantages. We have discussed in the revised manuiscipt, as “Doped catalysts can be synthesized via several methods, each with its own advantages. The sol-gel method, which involves hydrolyzing and condensing metal alkoxides or salts to form a gel that is then calcined, ensures homogeneous mixing and precise control (e.g., Ru3MoCeOx). Hydrothermal/solvothermal synthesis, which reacts metal precursors in a high-temperature/pressure solvent, produces highly crystalline materials with atomic-level doping (e.g., Ta0.1Tm0.1Ir0.8O2-δ). Co-precipitation, which precipitates metal ions together, is simple and cost-effective, yielding uniform mixed metal oxides (e.g., CoIr-CLEO). Impregnation, which loads metal precursors onto a support, enhances stability and activity by loading active metals onto supports (e.g., Pt0.1La0.1-IrO2@NC). Electrochemical methods, which deposit metal ions onto a substrate, create active and stable catalysts with controlled morphologies (e.g., Co0.2Zn0.8@RuO2).”.
- Is nitrogen-doped carbon considered a doping strategy, and are there other doped materials that could potentially be used in the OER? (doi.org/10.1039/d3ey00235g)
Answer: Thank you for raising this insightful question. We have carefully reviewed the literature you recommended and found it to be very helpful. Additionally, we discovered studies on the composite of carbon materials (CNT) with RuO2 (Composites Part B: Engineering 2022, 242, 110013.), which provided further insights. Although nitrogen (N) is not directly doped into RuO2 or IrO2, and this approach differs from the main catalysts summarized in our review, we are inspired by the literature you recommended. We believe that combining RuO2 or IrO2 with M-N-C composites could be a promising strategy to enhance catalyst performance. Therefore, we have included this article in the revised manuscript, which is referenced as [78]. We have added this part in the revised version as “Additionally, inspired by the literature on RuO2 composites with carbon materials [77, 78], exploring the potential of combining RuO2 or IrO2 with M-N-C composites ap-pears promising for enhancing catalyst performance.”.
- Various characterizations were mentioned in the Summary section. A detailed discussion can be added on how these techniques can be used together to better investigate the doped materials and their electrochemical properties.
Answer: Many thanks for this suggestion. The combined use of various characterization techniques can provide a comprehensive understanding of the doped materials and their electrochemical properties. For example, in the study of the Cr0.6Ru0.4O2 catalyst for acidic OER (Journal of the American Chemical Society 2024, 146, (46), 32049-32058.), in-situ Raman spectroscopy, in-situ ATR-SEIRAS, XPS, XAS, DFT calculations, and methanol molecular probe experiments were used together. In the revised manuscript as “For example, in the study of the Cr0.6Ru0.4O2 catalyst for acidic OER, in-situ Raman spectroscopy, in-situ ATR-SEIRAS, XPS, XAS, and DFT calculations were used together [34]. These techniques showed that Cr doping optimizes the Ru-O covalency, enhances the catalyst stability and activity, and promotes the direct oxygen radical coupling mechanism for efficient OER in acidic conditions.”.
- What is 2.2.3 section?
Answer: Thank you for pointing out our negligence. We have corrected “2.2.3” as “2.3” in the revised version.
Round 2
Reviewer 2 Report
Comments and Suggestions for Authors
The authors have addressed my questions. No additional comments are needed.